# Improved Peripheral and Hepatic Insulin Sensitivity after Lifestyle Interventions in Type 2 Diabetes Is Associated with Specific Metabolomic and Lipidomic Signatures in Skeletal Muscle and Plasma

**DOI:** 10.3390/metabo11120834

**Published:** 2021-12-03

**Authors:** Elin Chorell, Julia Otten, Andreas Stomby, Mats Ryberg, Maria Waling, Jon Hauksson, Michael Svensson, Tommy Olsson

**Affiliations:** 1Department of Public Health and Clinical Medicine, Umeå University, 901 87 Umeå, Sweden; julia.otten@umu.se (J.O.); andreas.stomby@umu.se (A.S.); mats.ryberg@umu.se (M.R.); tommy.g.olsson@umu.se (T.O.); 2Department of Food, Nutrition and Culinary Science, Umeå University, 901 87 Umeå, Sweden; Maria.waling@umu.se; 3Department of Radiation Sciences, Umeå University, 901 87 Umeå, Sweden; Jon.hauksson@regionvasterbotten.se; 4Department of Community Medicine and Rehabilitation, Section of Sports Medicine, Umeå University, 901 87 Umeå, Sweden; michael.svensson@umu.se

**Keywords:** exercise training, diet, type 2 diabetes, hepatic insulin sensitivity (hIS), peripheral insulin sensitivity (pIS), skeletal muscle, branched-chain amino acids (BCAA), diacylglycerol (DAG), ectopic fat

## Abstract

Lifestyle interventions with weight loss can improve insulin sensitivity in type 2 diabetes (T2D), but mechanisms are unclear. We explored circulating and skeletal muscle metabolite signatures of altered peripheral (pIS) and hepatic insulin sensitivity (hIS) in overweight and obese T2D individuals that were randomly assigned a 12-week Paleolithic-type diet with (diet-ex, *n* = 13) or without (diet, *n* = 13) supervised exercise. Baseline and post-intervention measures included: mass spectrometry-based metabolomics and lipidomics of skeletal muscle and plasma; pIS and hIS; ectopic lipid deposits in the liver and skeletal muscle; and skeletal muscle fat oxidation rate. Both groups lowered BMI and total % fat mass and increased their pIS. Only the diet-group improved hIS and reduced ectopic lipids in the liver and muscle. The combined improvement in pIS and hIS in the diet-group were associated with decreases in muscle and circulating branched-chain amino acid (BCAA) metabolites, specifically valine. Improved pIS with diet-ex was instead linked to increased diacylglycerol (34:2) and triacylglycerol (56:0) and decreased phosphatidylcholine (34:3) in muscle coupled with improved muscle fat oxidation rate. This suggests a tissue crosstalk involving BCAA-metabolites after diet intervention with improved pIS and hIS, reflecting reduced lipid influx. Increased skeletal muscle lipid utilization with exercise may prevent specific lipid accumulation at sites that perturb insulin signaling.

## 1. Introduction

The global burden of type 2 diabetes (T2D) continues to rise in conjunction with obesity, sedentary lifestyle, and energy-dense diets [1]. Though the underlying mechanisms remain unclear, T2D remission is possible via lifestyle changes with a success rate that is highly associated with disease duration [2,3] and the ability to reduce tissue fat [4].

In obesity, when the buffering capacity of adipose tissue to store excess fats is impaired, non-adipose tissues, such as muscle, liver, and pancreas, tend to accumulate lipids (ectopic lipids). This may induce a state of “cellular toxicity” and contribute to insulin resistance (IR) and β-cell dysfunction in T2D. Lipid intermediates, such as diacylglycerols (DAGs) and ceramides, are widely thought to be “lipotoxic” and the root cause of muscle IR [5], whereas triacylglycerols (TAGs) are considered more inert and not to be direct mediators of IR and T2D. In contrast, endurance-trained athletes have elevated muscle lipids despite being highly insulin sensitive [6]. This phenomenon can be explained in part by differences in lipid utilization and sub-cellular compartmentalization of metabolic by-products in the muscle [5] and the buffering capacity and insulin sensitivity in other tissues, such as the liver [7]. Oxidation of branched-chain amino acids (BCAAs) may play a key role in this phenomenon, including a suggested crosstalk between tissues [8]. During an obesity-driven decline in BCAA oxidation in adipose tissue, the oxidation of BCAAs has thus been suggested to be shuttled towards liver and skeletal muscle [9]. The increased BCAA influx may compete with lipid oxidation and lead to the accumulation of incompletely oxidized lipids, i.e., toxic lipid intermediates [10].

Intervention studies are key to explore causal links between improved insulin sensitivity and specific lipid intermediates and BCAA metabolism. We have therefore studied the effects of diet with and without added physical exercise on these putative mediators’ role in lifestyle interventions in people with T2D.

## 2. Results

### 2.1. Clinical Data

The baseline measures and intervention effects on anthropometric data, body composition, insulin sensitivity and ectopic fat for the full study have been published previously [11]. The sub-cohort for this study is shown in Table 1. At baseline, there were no significant differences between treatment groups regarding anthropometry, insulin sensitivity, ^1^HMRS-measured ectopic fat, or DXA-derived body composition. After intervention, both study groups had a significantly reduced BMI, sagittal abdominal diameter and % fat mass, decreased HbA1c, and improved pIS (*p* < 0.004). Only the diet-ex group had an improved VO_2_max and reduced C-reactive protein (*p* < 0.05). In contrast, only the diet group had increased hIS and reduced liver fat, skeletal muscle fat, and lean mass (*p* < 0.01). We observed no significant difference between the groups after intervention in regard to hIS and pIS, the HMRS-measured ectopic fat, or DXA-derived body composition.

### 2.2. Metabolomics and Lipidomics

No outliers were detected in the initial PCA inspection of the dataset (data not shown). In both treatment groups, we found significantly altered skeletal muscle (OPLS-EP, CV-ANOVA *p* < 0.05, Figure 1) and plasma (OPLS-EP, CV-ANOVA *p* < 0.05, Figure 2) metabolite and lipid signatures following the 12-week intervention. If not stated otherwise, all discussed changes in metabolite and lipid levels are significant according to the criteria listed in the Methods section. A complete list of all detected metabolites and lipids included in the multivariate models are presented in Appendix A.

#### 2.2.1. Skeletal Muscle Metabolomics and Lipidomics

In the diet group, skeletal muscle BCAA valine was reduced post-intervention (Figure 1a,c) with no changes in isoleucine/leucine (Figure 1a), with the latter not being completely separated by chromatography and, thus, not able to be separately quantified. In addition, the diet group had lower levels of glucose-6-phosphate, the cholesterol intermediate squalene [12], and sarcosine, a proteolytic marker [13] post-intervention, but these were not altered in the diet-ex group (Figure 1b). BCAAs were not changed in the diet-ex group, but other amino acids (lysine, citrulline and arginine) were reduced post-intervention, as well as nucleotide (inosine), myo-inositol, and hexoseamines acetylglucoseamine and acetylfructoseamine (Figure 1b), which were not altered in the diet group.

The diet group had an overall decrease in muscle lipids (Figure 1d,f). More specifically, we noted a decrease in ceramides, specifically cer (d18:1/20:4), cer (d18:1/23:0), cer (d18:0/24:3), cer (d18:1/24:1), and galactose-cer (d18:1/23:1); DAGs, specifically DAG (30:0), DAG (34:2), DAG (34:1), and DAG (38:4); and numerous phosphatidylcholines (PCs), sphingomyelins (SMs), and TAGs (Figure 1d,f). We found that all TAGs that decreased in the diet group were polyunsaturated (i.e., had more than five double bonds in their combined fatty acyls: Figure 1f).

The diet-ex group exhibited a diverse intervention response in their skeletal muscle lipids, with a decrease in cer (d18:1/23:0), cer (d18:1/24:3), cer (d18:1/24:0), galactose-cer (d18:1/23:1), SM (d18:1/16:0), SM (d18:1/18:1), SM (d181:18:0), SM (d18:1/24:2), DAG (40:6), and numerous PCs and TAGs (Figure 1e,g). In addition, the diet-ex group had increased DAG (34:2), DAG (34:1), PC (30:0) (Figure 1e) and saturated and monounsaturated (SAT/MUFA) TAGs (i.e., less than two double bonds in their combined fatty acyls; Figure 1g). Similarly, as observed in the diet group, polyunsaturated fatty acid (PUFA)-TAGs were decreased in the diet-ex group (i.e., more than five double bonds in their fatty acyls).

We measured the levels of cardiolipins, a mitochondrial-specific lipid and established marker of mitochondrial content [14], in a sample subset from individuals with remaining skeletal muscle extract (diet *n* = 7, diet-ex *n* = 4). All subjects in the diet-ex group increased their total cardiolipin content, whereas the diet group had a mixed response (Figure 1h). A complete list of all detected cardiolipin species and total content is presented in Appendix A. No statistics were calculated on the cardiolipin data due to the low number of samples analysed in each group.

#### 2.2.2. Plasma Metabolomics and Lipidomics

The diet group had a post-intervention decrease in circulating BCAA valine and its catabolic intermediate 3-hydroxyisobutyric acid (3-HIB; Figure 2a), with no change in the diet-ex group (Figure 2b). The relative concentrations of plasma valine and 3-HIB in each intervention group are shown in Figure 2e,f. In addition to valine, the diet group had reduced levels of several other amino acids, including alanine, threonine, proline, tryptophan, tyrosine, and phenylalanine, along with a decrease in sugars sorbose, galactose, maltose, fructose, cellobiose, fucose, and glucose (Figure 2a). The diet group also increased their C6-C20-acylcarnitines and the ketone body 3-hydroxybutyric acid (3-HB), whereas lactic acid and the glycine-conjugated bile acids deoxycholic acid-glycine conjugate (G-DCA) and cholic acid-glycine conjugate (G-CA) decreased (Figure 2a).

The diet-ex group did not have altered levels of plasma BCAAs or amino acids post-intervention, with the exception of a decrease in tyrosine (Figure 2b). Similarly, we found no change in the circulating levels of sugars in the diet-ex group except for a decrease in maltose and cellobiose (Figure 2b).

We also observed an increase in PUFA-fatty acids C18:3-n3, C18:2-n6, C18:1-n9, C20:4-n6, and C22:6-n3 in the diet group, with no change in their saturated fatty acids (Figure 2c). A few circulating lipids were altered in the diet group: a decrease in DAG (32:3), DAG (38:5), PC (32:1), PC (38:2), and lysophosphatidylcholine (LPC) (14:0), and an increase in LPC (18:0). Numerous TAG species decreased in the diet group (Figure 2c); these changes were even more notable in the diet-ex group (Figure 2d).

In contrast, the diet-ex group exhibited a profound alteration in their circulating lipids post-intervention (Figure 2d), which included a decrease in saturated medium-long fatty acids C9:0, C10:0, C14:0, and C18:0. Furthermore, we found an increase in their PUFA-fatty acids C18:2-n6 and C22:6-n3a, and a decrease in DAG (32:3), DAG (36:4), DAG (36:0), DAG (30:0), and DAG (38:5) along with numerous PCs, cholesterol, and TAGs (Figure 2d). In addition, the diet-ex group had increased levels of specific lysophospholipids, specifically LPC (18:1), LPC (18:0), LPC (18:2), and LPC (20:4), whereas LPC (14:0) was decreased.

### 2.3. Intervention Response in Skeletal Muscle Metabolites and Lipids That Describe Improved Tissue-Specific Insulin Sensitivity

The pIS increased significantly in both intervention groups (Figure 3a), whereas hIS only increased significantly in the diet group (Figure 3b). For both intervention groups, we detected a skeletal muscle metabolite/lipid signature that was significantly associated with improved pIS (OPLS, CV-ANOVA *p* < 0.05, Figure 3c). We found no link between the altered skeletal muscle lipid and metabolite signatures that could describe intervention-related changes in hIS.

Improved pIS in the diet group was associated with a decrease in the skeletal muscle BCAAs valine and isoleucine/leucine, glycerol-2-phosphate, and other amino-containing compounds, such as sarcosine, threonine, and alanine, and an increase in myo-inositol levels (Figure 3c). We found no association between improved pIS and skeletal muscle BCAA levels in the diet-ex group (Figure 3c). Instead, improved pIS in the diet-ex group was associated with increases in TAG (56:0) and DAG (34:2) and decreases in PC (34:4) and numerous amino-containing compounds, specifically N-acetylmannoseamine, N-acetylglucoseamine, inosine, ornithine, citrulline asymmetric dimethylarginine, arginine, sarcosine, and creatinine. We also found that improved pIS in the diet-ex group was associated with lower myo-inositol, inosine, and chlorogenic acid levels (Figure 3c). We observed that the skeletal muscle fat oxidation rate during exercise were associated with an increase in SAT/MUFA-TAGs and lowering of PUFA-TAGs with the intervention (Figure 4, OPLS, CV-ANOVA *p* < 0.05).

### 2.4. Intervention Response in Circulating Metabolites and Lipids That Describe Improved Tissue-Specific Insulin Senstitivty

We observed no significant association between improved pIS and alterations in circulating metabolites and lipids (data not shown). The diet group had improved hIS (Figure 3d). This study group had a significant association between improved hIS and an altered circulating metabolite signature (OPLS, CV-ANOVA *p* = 0.04, Figure 3d). In addition, the improved hIS in the diet group was associated with a decrease in circulating amino acids and BCAA metabolites (i.e., isoleucine, leucine, valine, and their catabolic intermediates 3-HIB and ketoleucine), as well as decreases in lactic acid and circulating sugars rhamnose, 1,5-anhydroglucitol, and laminaribose. We also found that the improved hIS was associated with increases in some fatty acids (i.e., C22:6-n3, C16:0, and C20:3-n3; Figure 3d).

## 3. Discussion

Insulin sensitivity in the skeletal muscle and liver is a key target for interventions in type 2 diabetes (T2D). A novel finding in this study is that combined improvement in pIS and hIS with a diet intervention in patients with T2D was associated with decreased skeletal muscle and circulating levels of valine and its circulating catabolic intermediate 3-HIB. In contrast, a combined diet and exercise intervention that improved pIS and skeletal muscle fat oxidation, without improving hIS, was associated with altered skeletal muscle lipids. Our results suggest that improvements in pIS in T2D are mediated by different mechanisms if accompanied by improvements in hIS or muscle fat oxidation capacity (Figure 5).

Our findings suggest that BCAA-related pathways, and valine specifically, are involved in mediating the combined improvement in hIS and pIS in overweight T2D patients. Recent mechanistic data support a BCAA crosstalk mechanism between the skeletal muscle and liver, in which the liver removes excess nitrogen (NH3) generated from obesity-associated increases in BCAA transamination in the skeletal muscle [18]. An increased transamination has thus been shown to lead to toxic lipid intermediates, followed by impaired glucose uptake.

Notably, 3-HIB is obtained from valine catabolism and has been shown to activate trans-endothelial fatty acid transport and promote lipid accumulation and IR in muscle via incompletely esterified intermediates, such as DAGs, that can disrupt insulin signaling [15,19]. After the initial BCAA oxidation step, the BCAA metabolites are trapped inside of the mitochondria, with the exception of 3-HIB, which can pass through membranes and participate in tissue crosstalk. Studies in the 1980s revealed 3-HIB to be a carrier of the “glucogenic potential” of valine, as 3-HIB can be converted to glucose in the liver [20], supporting its paracrine effects. Our finding of decreased circulating 3-HIB and muscle valine levels in the diet group accompanied by decreases in ^1^HMRS-measured lipids and lipidomics lipid subtypes in muscle suggests lower fatty acid transport into the muscle via this mechanism, that can be mediated by increased hIS. This is supported by an increase in circulating acylcarnitines in the diet group, which may reflect decreases in lipid depots [16]. Importantly, we did not find any association between increased circulating acylcarnitines and improved tissue-specific insulin sensitivity. As acylcarnitines are needed to shuttle fatty acids towards mitochondrial beta-oxidation, increases in circulating acylcarnitines may reflect tissue-detoxification of non-oxidized fatty acids due to decreased ectopic fat without improving fat utilization. Therefore, analyses of skeletal muscle acylcarnitine levels are of interest in future intervention studies.

We did not find a reduction in the ^1^HMRS-measured fats in the liver and muscle in the diet-ex group despite an increased muscle fat oxidation capacity and weight loss. This implies increased fat utilization in the skeletal muscle. Both individuals with T2D and insulin-sensitive endurance-trained athletes have elevated muscle lipids [6], which in part can be explained by differences in the lipid utilization capacity and sub-cellular compartmentalization of lipids.

Mitochondrial lipids have been shown to be associated with high turnover rates and insulin sensitivity, whereas plasma membrane lipids have been associated with a low turnover rate and IR [5,21]. In addition, DAGs have been shown to disrupt the insulin signalling cascade via inhibition of tissue-specific kinases [19]. However, this may only apply to DAGs that accumulate at the plasma membrane [5], not at the endoplasmic reticulum/Golgi apparatus, which may provide lipids to the mitochondria in association with increased lipid utilization and mitochondrial biogenesis [22]. In line with this, the improvement in pIS after a combined diet and exercise intervention was associated with an increase in skeletal muscle cardiolipin, a validated marker of mitochondrial content [14], and an increased skeletal fat oxidation capacity coupled with a specific increase in DAG (34:2). Our findings are supported by results from an exercise intervention in obese subjects without diabetes, in whom mitochondrial content and mitochondrial fat oxidation improved without alterations in the total muscle lipid content [17]. The increase in skeletal muscle hexoseamines in our study also fits with increased fat utilization as these acetylated sugar amines are involved in post-translational protein modifications that serve as a nutrient signal, increasing glucose uptake and glycolytic enzyme activity in the skeletal muscle [23]. Whether this increased lipid utilization in skeletal muscle counteracts the initial depletion of ectopic lipids and putative improvements in hIS warrants further investigation.

Skeletal muscle ceramides were decreased in both intervention groups, but this was not associated with improved pIS, as increased ceramide have been shown to be associated with established IR and T2D [24], and may possibly not be reversed by short-term interventions in T2D. Long-term lifestyle interventions are needed to verify/refute this assumption.

TAGs may have multiple roles on a tissue level in metabolic dysfunction. For both intervention groups, we observed a decrease in skeletal muscle PUFA-TAGs. Although TAGs are considered inert and not involved in perturbing insulin signalling mechanisms, they may play an indirect antioxidant role by harbouring PUFAs to protect against cellular peroxidation [25]. In addition, the diet-ex group increased their SAT/MUFA-TAGs in muscle, and an increase in SAT-TAG (56:0) was associated with improved pIS. We also showed that increased skeletal muscle SAT/MUFA-TAGs were associated with higher skeletal muscle fat oxidation. Further studies of the degree of TAG saturation in skeletal muscle and its link to lipid homeostasis and lipotoxicity are needed in both prediabetes and overt T2D. A previous publication from our study group shows that the applied Palaeolithic-type diet increased dietary PUFA content, which was associated with lower levels of circulating TAGs independent of weight loss [26]. However, the study by Martensson et al. [26] could not conclude any difference in dietary adherence between intervention group. Of relevance is that increases in dietary PUFA, such as in Mediterranean diets, have previously shown similar increases in insulin sensitivity after 12 weeks of diet [27].

Our study subjects were well-characterized from a metabolic perspective, using hyperinsulinemic euglycemic clamps to estimate both hIS and pIS, together with ^1^HMRS-measured lipids in the liver and skeletal muscle. However, a weakness of this study is the limited sample size, especially for the lipid analyses and analyses of mitochondrial content. Another limitation of the study is potentially confounding effects of medications and differences in baseline insulin sensitivity, which should be considered in future studies. Still, our data suggest different adaptive mechanisms underlying improved tissue-specific insulin sensitivity with lifestyle interventions that are amenable to further studies.

## 4. Materials and Methods

### 4.1. Research Design and Sampling Procedure

This study is a secondary analysis of an intervention study, eligibility criteria have been presented previously [11,28]. Briefly, we included 32 overweight and obese (BMI 25–40 kg/m^2^) weight-stable (i.e., <5% weight loss) men and postmenopausal women with T2D. The participants were instructed to eat a Palaeolithic-type diet for 12 weeks. The diet was consumed ad libitum and based on lean meat, fish, nuts, and vegetables. Dairy products, cereals, refined fats and sugars, and salt were excluded. All participants received (group-wise) guidance from a dietician throughout the intervention. A previous publication showed that this Palaeolithic-type diet significantly altered the participants carbohydrate, protein, and fat intake by means of a decreased carbohydrate and saturated fat intake and an increased intake of protein and mono- and polyunsaturated fats [11]. Prior to randomization, all study participants were advised to perform 30 min of moderate exercise daily according to current diabetes treatment guidelines. The subjects were then randomized to supervised exercise training for 3 h per week (diet-ex group) or to maintaining the standard care exercise recommendations (diet group). The diet-ex group underwent three supervised exercise sessions/week, including both resistance and aerobic exercise, with an experienced personal trainer in accordance with the guidelines of the American College of Sports Medicine [29].

Blood samples and muscle biopsies were collected at baseline and after 12 weeks of intervention. Plasma samples were collected after an overnight fast according to standardized routines with a minimum at-bench time and immediately stored in −80 °C. Muscle biopsies were obtained from the lateral lower portion of the *m. vastus lateralis* under local anaesthesia (Carbocain and Adrenalin 5 mg/mL, AstraZeneca, Södertälje, Sweden). A conchotome was used to collect all muscle tissue samples via a small (2–3 cm) incision through the skin and fascia. Visible fat, connective tissue, and blood clots in the biopsy sample were immediately removed under a dissection microscope. Each muscle sample was rapidly frozen in liquid propane combined with liquid N2 (−160 °C) and stored in a freezer at −80 °C until further preparation and analysis.

This study included a subset of subjects with available muscle and plasma samples at both baseline and post-intervention (Table 1). Twenty-six subjects (diet-ex *n* = 13 and diet *n* = 13) with T2D an HbA1c between 6.5% and 10.8% (47–94 mmol/mol) and treated with diet and/or metformin were included. Exclusion criteria were antidiabetic drugs other than metformin, use of beta-blockers, blood pressure > 160/100 mmHg, macroalbuminuria, cardiovascular disease, and higher amounts of training (e.g., moderate endurance training five times a week, resistance training every other week). This study is registered as a clinical trial (NCTT01513798) at ClinicalTrials.gov.

### 4.2. Body Composition, Tissue-Specific Insulin Sensitivity, Liver and Soleus Fat, Vo_2_max, and Fat Oxidation Capacity

Measures of body composition, hepatic (hIS) and peripheral insulin sensitivity (pIS), and liver and skeletal muscle ectopic fat for the complete study population were reported previously [28]. Briefly, body composition, including subtotal (excluding the head) fat mass and fat-free soft tissue mass, was measured by dual-energy X-ray absorptiometry (Lunar Prodigy X-ray Tube Housing Assembly, Brand BX-1L, Model 8743; GE Medical Systems, Madison, WI, USA). Sagittal abdominal diameter was measured as the height of the abdomen at the umbilical level when the subject was lying down on a flat surface with their legs straight [30]. Tissue-specific insulin sensitivity was measured using the hyperinsulinemic-euglycemic clamp technique combined with [6,6-^2^H] glucose infusion as described previously [28]. The pIS was measured as the rate of disappearance during the hyperinsulinemic clamp and hIS as the suppression of endogenous glucose production. Both hIS and pIS were normalized for plasma insulin during the clamp [28]. Liver and soleus muscle lipids were analysed by proton magnetic resonance spectroscopy (^1^HMRS). A graded exercise testing protocol was applied to determine the maximal oxygen uptake (VO_2_max), which included ergometer cycling without any resistance for a 3 min warm-up, followed by increased resistance, corresponding to 10–20 W depending on each individual’s fitness level, every minute until volitional exhaustion or reaching a plateau, or levelling-off, of VO_2_. The mean rate of fat oxidation was calculated from the measured levels of VCO_2_ and VO_2_ [31] during a submaximal ergometer cycling (Monark, 894, Monark Exercise AB, Vansbro, Sweden) session measured at 55% of maximal aerobic power. The fat oxidation rate was calculated using the stoichiometric equation by Jeukendrup and Wallis [31], which is mainly considered to reflect the muscle fat oxidation capacity. All analyses of air flow and respiratory gases were performed with a Jaeger Oxycon Pro system (Erich Jaeger GmbH, Hoechberg, Germany).

### 4.3. Mass Spectrometry-Based Metabolomics Analysis

#### 4.3.1. Sample Preparation

Sample preparations and analytical run order were designed to circumvent methodological biases interfering with the interpretation of results [32]. Samples from the same individual were prepared and analysed in close connection while keeping the internal sample order randomized. Analytical batches were balanced in terms of treatment group and quality control (QC) samples were continuously analysed. A detailed description of sample preparation, internal standards, drift removal, and data normalization are provided in the Appendix A. Briefly, plasma samples were prepared according to A et al. [33] using 90/10, *v*/*v* methanol: water extraction including internal standards for metabolomics, and a 70/30, *v*/*v* chloroform: methanol extraction for lipidomics [34]. Muscle tissues were prepared according to Gullberg et al. [35] and A et al. [33] using 80/20, *v*/*v* methanol:water for metabolomics analysis and 70/30, *v*/*v* chloroform: methanol extraction for lipidomics. Plasma samples were subjected to both GC-MS metabolomics, LC-MS metabolomics, and lipidomics analyses. Due to limited amounts of muscle tissue being available, samples from all subjects were analysed on the GC-MS platform for metabolomics and a subset of 12 samples with a sufficient amount of tissue were also analysed on the LC-MS lipidomics platform (diet = 8 and diet-ex = 4).

#### 4.3.2. Data Processing

The GC-MS data were processed using an in-house MATLAB script, R2016a (The MathWorks, Inc., Natick, MA, USA). The LC-MS data were processed using Agilent Masshunter Profinder version B.08.00 (Agilent Technologies Inc., Santa Clara, CA, USA). Putative metabolites were extracted using unique mass channels, retention indices (GC-MS data), MS-MS spectra (LC-MS dat) and matched to our in-house mass spectral library at the Swedish Metabolomics Centre (www.swedishmetabolomicscentre.se, 10 October 2021). Labelled internal standards were used for alignment and normalization. Extensive filtering was performed to remove noise and entities with poor quality (e.g., suffering from peak broadening, ion suppression, or non-Gaussian peaks), and only unique spectral profiles with a relative standard deviation (RSD) < 40% calculated from QC samples were included in sample comparison modelling. Criteria set by the Human Metabolome Database (www.HMDB.ca, 15 August 2021) were used to assign extracted components to different compound classes (e.g., amino acids and derivatives, BCAAs, carbohydrates, lipid subtypes, or no class). All lipids were annotated according to standard lipid nomenclature set by *Lipid Maps Lipidomics Gateway* (lipidmaps.org).

#### 4.3.3. Statistical Analysis and Bioinformatics

The statistical analysis included both univariate and multivariate analyses and was carried out using MATLAB R2016a (The MathWorks, Natick, MA, USA) and SIMCA 16.0.0 software (Sartorius, Umeå, Sweden). The non-parametric Wilcoxon–Mann–Whitney rank sum test was used for univariate comparisons of changes in variables (anthropometric data, clinical markers, and metabolites) among different diagnosis groups at different sample collection time points.

For the multivariate analysis, the principal component analysis (PCA) was used to evaluate groupings, outliers, and trends. Next, orthogonal partial least squares (OPLS) analyses were applied to study the relationships between metabolomic/lipidomic profiles and relevant end points, such as treatment and tissue-specific insulin sensitivity. Furthermore, we applied a variant of OPLS, OPLS-effect projections [32], in which each subject’s baseline metabolic profiles were subtracted from its post-intervention metabolite/lipid profile. By using this approach, the treatment effects on the metabolomics/lipidomic profile can be evaluated and the influence from instrumental drift, inter-individual variation, and multiple testing minimized. The above models were validated based on analysis of variance of the cross-validated OPLS scores (CV-ANOVA) for significance testing. A metabolite was considered to be significantly altered based on a significant univariate *p*-value and a jack knifing-based confidence interval from OPLS models [36]. A 95% significance level was applied throughout this work.

## 5. Conclusions

In conclusion, we found that improved pIS via dietary intervention is associated with reduced muscle valine and its catabolic intermediate 3-HIB, in parallel with improved hIS and decreased lipid content in the skeletal muscle and liver. In contrast, improved pIS after a diet intervention combined with supervised exercise training alters muscle lipids and other nutrient metabolites indicative of increased energy metabolism. This is accompanied by an increased muscle fat oxidation rate and an increase in mitochondrial content without changes in liver or muscle fat content. Our results highlight the need for more mechanistic studies of specific biochemical pathways involved in tissue-specific IR in T2D.

## Figures and Tables

**Figure 1 metabolites-11-00834-f001:**
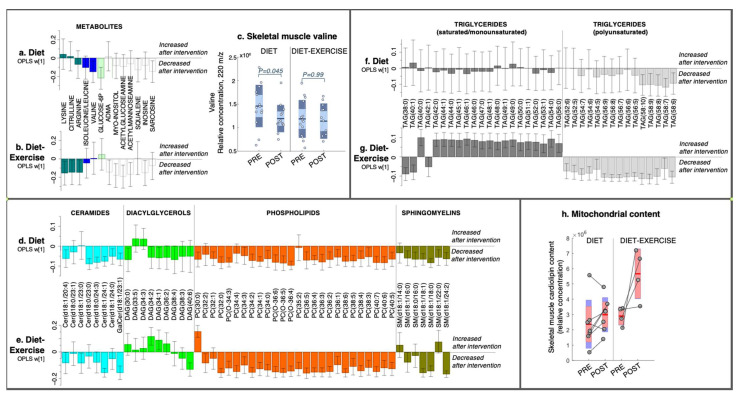
Intervention-specific multivariate responses in skeletal muscle. (**a**,**b**) Metabolites, (**d**,**e**) lipids, i.e., phospholipids and lipid intermediates, and (**f**,**g**) triacylglycerides from a 12-week diet intervention or diet combined with exercise. Coefficients were calculated from intervention-specific OPLS-EP models based on the delta change in metabolites and lipids in each individual (post-intervention—basal). Y-axis values describe the multivariate response for each metabolite and lipid (w [1]). 95% confidence levels are given. Only metabolites and lipids significant in at least one intervention group are shown. All detected metabolites and lipids are found in Appendix A. (**c**) Individual levels of the relative abundance of valine in the skeletal muscle. (**h**) Raw data and the relative abundance of cardiolipin in the total skeletal muscle and an estimate of mitochondrial content. Each individual’s pre- and post-sample levels are connected with a line. ADMA, asymmetric-dimethylarginine; Cer, ceramide; DAG, diacylglycerol; PC, phosphatidylcholine; SM, sphingomyelin; TAG, triacylglycerol. The numbers of carbons and double bonds are described in parentheses for each lipid species, e.g., TAG (58:10) is a triacylglycerol in which its three fatty acyl groups together contain 58 carbons and 10 double bonds. The exact fatty acyl composition is shown for Cers and SMs in parentheses.

**Figure 2 metabolites-11-00834-f002:**
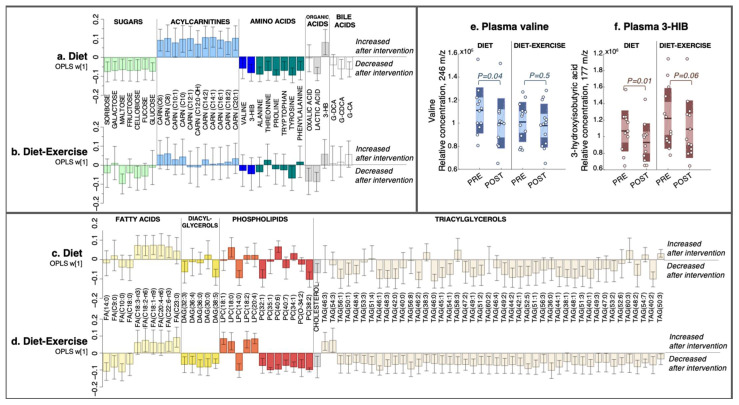
The multivariate response after 12 weeks of diet only and diet combined with supervised exercise. (**a**,**b**) Plasma metabolites, (**c**,**d**) lipids. Coefficients were calculated from an intervention-specific OPLS-EP model based on the delta change in metabolites and lipids in each individual (post-intervention—basal). Y-axis values describe the multivariate response for each metabolite and lipid (w [1]). 95% confidence levels are given. Only metabolites and lipids significant in at least one intervention group are shown. All detected metabolites and lipids are found in Appendix A. (**e**) Individual data for the relative abundance of plasma valine and (**f**) its catabolic intermediate 3-hydroxyisobutyric acid (3-HIB). FA, fatty acid; DAG, diacylglycerol; LPC, lysophosphatidylcholine; PC, phosphatidylcholine; CARN, acylcarnitine; TAG, triacylglycerol; 3-HIB, 3-hydroxyisobutyric acid; 3-HB, 3-hydroxybutyric acid; G-DCA, deoxycholic acid-glycine conjugate; G-CDCA, chenodeoxycholic acid-glycine conjugate; G-CA, cholic acid-glycine conjugate.

**Figure 3 metabolites-11-00834-f003:**
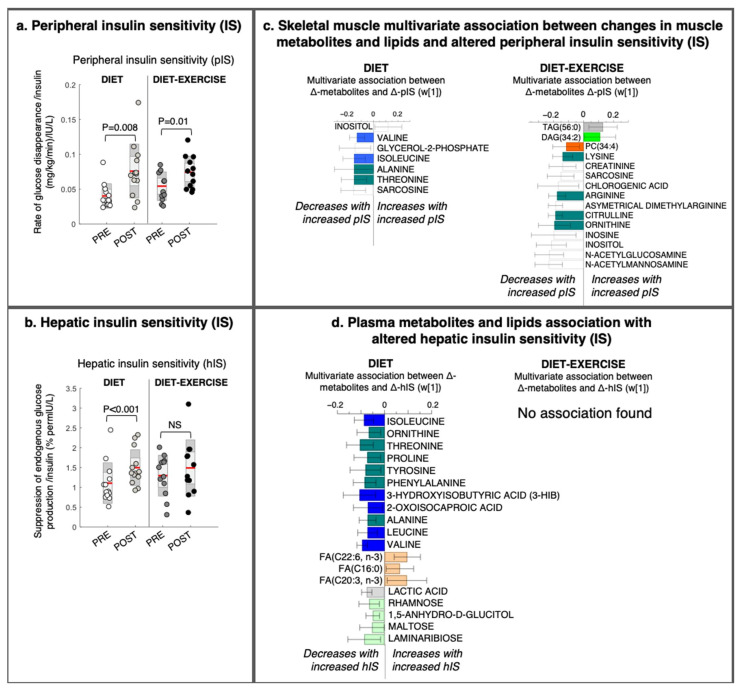
Changes in tissue-specific insulin sensitivity and its multivariate association with changes in metabolites and lipids from a lifestyle intervention with diet or diet in combination with exercise (diet-exercise) in subjects with type 2 diabetes. (**a**) Peripheral and (**b**) hepatic insulin sensitivity measured during a hyperinsulinemic euglycemic clamp at baseline and after 12 weeks of diet or diet-exercise. (**c**) Intervention-specific multivariate association between changes in peripheral insulin sensitivity and skeletal muscle lipids and metabolites (OPLS weights (w [1]), CV-ANOVA *p* < 0.05). (**d**) Intervention-specific multivariate association between changes in hepatic insulin sensitivity and changes in circulating lipids and metabolites (OPLS weights (w [1]), CV-ANOVA *p* < 0.05). 95% confidence levels are given. No significant association was found between changes in hepatic insulin sensitivity and circulating lipids and metabolites in the diet-exercise group. PC, phosphatidylcholine; DAG, diacylglycerol; TAG, triacylglycerol; FA, fatty acid.

**Figure 4 metabolites-11-00834-f004:**
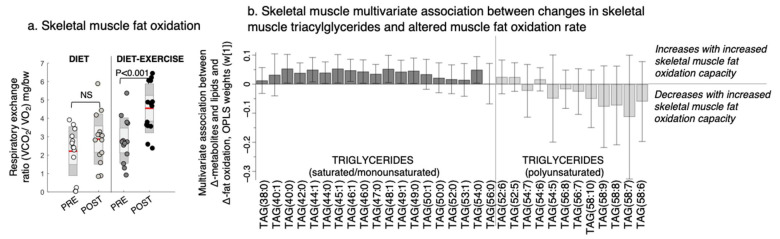
Changes in the skeletal muscle fat oxidation rate and its multivariate association with changes in triacylglycerols (TAGs) after a lifestyle intervention with diet or diet in combination with exercise (diet-exercise) in subjects with type 2 diabetes. (**a**) Skeletal muscle fat oxidation measured during submaximal exercise at baseline (pre) and after 12 weeks of lifestyle intervention (post). NS indicates non-significant difference. (**b**) Multivariate association between changes in the skeletal muscle fat oxidation capacity and skeletal muscle TAGs (OPLS weights (w [1]), CV-ANOVA *p* < 0.05).

**Figure 5 metabolites-11-00834-f005:**
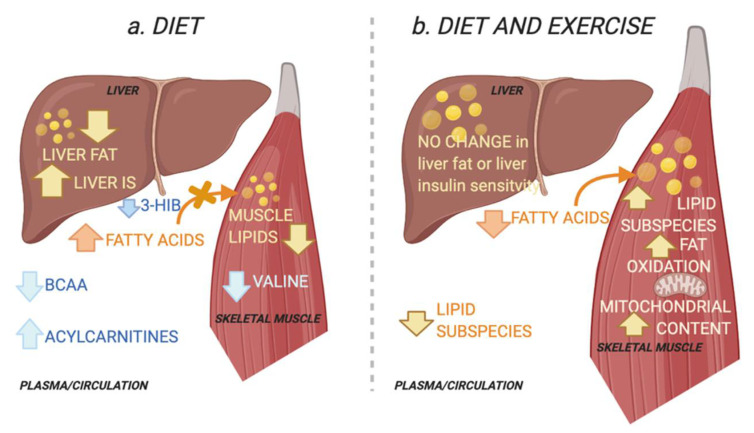
Schematic overview of the working hypothesis. (**a**) The diet group improved their hepatic insulin sensitivity (IS) and reduced ectopic lipids in the liver and muscle. Improved peripheral IS in the diet group was associated with reduced skeletal muscle and plasma levels of the branched chain amino acid (BCAA) valine and plasma levels of its catabolic intermediate 3-hydroxyisobutyric acid (3-HIB) in skeletal muscle. The latter increases transendothelial fatty acid influx [15]. This is associated with an increase in circulating acylcarnitines, congruent with a decrease in lipid depots [16]. Decreases in circulating BCAAs suggest increased hepatic BCAA oxidation as a result of increased hepatic IS and reduced liver fat [9]. (**b**) The combined diet and exercise group did not alter their hepatic IS or their total ectopic lipid levels in the liver and muscle. Instead, peripheral IS was associated with altered muscle lipids, i.e., increase in specific diacylglycerols and triacylglycerols. The diet-exercise group also had increased skeletal muscle fat oxidation capacity and mitochondrial content. This may reflect increased lipid utilization and a redistribution of lipids towards the more bioenergetic organelles and increased lipid utilization, instead of the plasma membrane where lipids may interfere with insulin signalling [5,17]. Created with BioRender.com (29 November 2021).

**Table 1 metabolites-11-00834-t001:** Subject characteristics.

	Diet Group		Diet-Exercise Group	
	PRE (Week 0)	POST (Week 12)	PRE (Week 0)	POST (Week 12)
Female subjects (male subject), *n*	12 (9)		14 (9)	
Age, years	58.3 ± 6.8		62.3 ± 4.5	
Diabetes duration, years	4.2 ± 3.2		4.5 ± 3.4	
Weight, kg	94.5 ± 14.2	86.0 ± 11.8 *	95.5 ± 17.5	89.0 ± 17.1 *
BMI, kg/m^2^	31.6 ± 3.0	28.7 ± 2.2 *	31.9 ± 3.7	29.6 ± 3.9 *
Sagittal abdominal diameter, cm	27.1 ± 3.4	23.2 ± 2.9 *	26.3 ± 3.7	23.7 ± 3.2 *
VO_2_ peak, ml/min	2303.6 ± 362.2	2265.3 ± 351.1	2270.5 ± 647.9	2428.0 ± 593.2 *
Waist circumference, cm	110.5 ± 9.8	100.6 ± 8.1 *	109.7 ± 10.9	102.5 ± 12.6 *
Lean mass	58.1 ± 10.9	55.5 ± 9.2 *	56.9 ± 13.0	57.0 ± 12.9
Fat mass, %	36.2 ± 6.1	26.8 ± 6.7 *	38.3 ± 5.0	33.5 ± 5.7 *
CRP, nmol/l	1.7 ± 1.3	1.3 ± 0.9	1.8 ± 1.3	1.2 ± 0.8 *
HbA1c, %	6.2 ± 1.7	5.2 ± 1.7 *	6.7 ± 2.4	5.3 ± 2.3 *
INSULIN SENSITIVITY	PRE (Week 0)	POST (Week 12)	PRE (Week 0)	POST (Week 12)
Hepatic insulin sensitivity ^a^	1.1 ± 0.5	1.5 ± 0.4 *	1.3 ± 0.5	1.3 ± 0.5
Peripheral insulin sensitivity ^b^	0.04 ± 0.02	0.08 ± 0.04 *	0.05 ± 0.02	0.07 ± 0.02 *
Insulin resistance estimate (HOMA-IR) ^c^	7.5 ± 2.4	4.0 ± 1.9 *	7.2 ± 3.0	3.6 ± 2.0 *
ECTOPIC FAT	PRE (Week 0)	POST (Week 12)	PRE (Week 0)	POST (Week 12)
Liver fat, %	18.1 ± 9.7	6.1 ± 5.9 *	15.3 ± 9.8	9.7 ± 8.3
Skeletal muscle fat, % in soleus	20.4 ± 8.5	11.9 ± 6.4 *	22.8 ± 14.8	19.8 ± 16.1

Data are presented as means ± SD unless otherwise noted. ^a^ Endogenous glucose production, mg kg^−1^min^−1^/basal insulin. ^b^ Rate of disappearance, mg kg^−1^ min^−1^/basal inulin. ^c^ Homeostatic model assessment for insulin resistance. ** p* < 0.05 within group (week 0 vs. week 12).

## Data Availability

The datasets generated during and/or analyzed during the current study are available from the corresponding author on reasonable request. The data are not publicly available due to data protection regulation law.

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
