# Peer review of "Improved Peripheral and Hepatic Insulin Sensitivity after Lifestyle Interventions in Type 2 Diabetes Is Associated with Specific Metabolomic and Lipidomic Signatures in Skeletal Muscle and Plasma"

_metabolites, 2021, doi:10.3390/metabo11120834_

Round 1
Reviewer 1 Report
Dear author,
the manuscript depicting the different patterns between a diet and a diet-assisted with exercise appears well written and clear in the presentation of the results.
Clearly, this study need to continue in the analyzing of the main pathways that regulate IS in the 2 groups.
I have only one concern:
- in table 1 the author should include the HbA1c in % for consistency of the measurements.
Author Response
We thank the reviewer for their encouraging comments. Validation of findings/ pathways are planned and ongoing. According to the reviewer’s comment, the HbA1c data in table 1are now presented in %.
Reviewer 2 Report
In this prospective interventional, comparative, cohort study, Chorell et al addressed the changes in BCAA-metabolites and lipids in skeletal muscle and circulating plasma following 12 weeks of paleolithic diet, either alone or in combination with supervised exercise training (3h/week, in addition to standard recommendations of physical activity in diabetes care) and their correlation with hepatic or systemic insulin sensitivity.
The study is well written and interesting to read. Furthermore, the methods used to analyze the metabolomics/lipidomics data are statistically sound. I have only minor concerns:
- As acknowledged by the Authors themselves, the sample size is quite limited (n=32; with 1:1 randomization ratio). However, I would suggest checking the percentage of women enrolled in each group (9%? Please see table 1) and make clear for the reader that they were all post-menopausal women. By limiting variations related to menstrual cycle, this would be a strength point of the study.
- Another limit of the study is that some differences can be found with respect to baseline features for the two patient’s groups (i.e., age, insulin sensitivity indices) and this may have biased, at least to some extent the observed results. Also, some patients may have been more compliant than others with respect to paleolithic diet, standard recommendations for physical activities and even metformin use. As optimal matching is quite difficult to occur in a small, randomized study like this, I would suggest strengthening the need for larger studies to corroborate the present metabolomics data.
- The considerable amelioration of insulin sensitivity after only 12 weeks of paleolithic diet (independently of additional exercise) is unsurprising in view of recent evidences of a protective role for Mediterranean diet foods against insulin resistance. Please refer to doi: 10.3390/nu12041066 and doi: 10.3390/nu12103042 as suitable references.
Author Response
In this prospective interventional, comparative, cohort study, Chorell et al addressed the changes in BCAA-metabolites and lipids in skeletal muscle and circulating plasma following 12 weeks of paleolithic diet, either alone or in combination with supervised exercise training (3h/week, in addition to standard recommendations of physical activity in diabetes care) and their correlation with hepatic or systemic insulin sensitivity.
The study is well written and interesting to read. Furthermore, the methods used to analyze the metabolomics/lipidomics data are statistically sound. I have only minor concerns:
As acknowledged by the Authors themselves, the sample size is quite limited (n=32; with 1:1 randomization ratio). However, I would suggest checking the percentage of women enrolled in each group (9%? Please see table 1) and make clear for the reader that they were all post-menopausal women. By limiting variations related to menstrual cycle, this would be a strength point of the study.
RESPONSE: We thank the reviewer for this valuable comment highlighting an error, which was also pointed out by reviewer 3 and addressed accordingly:
The presented distribution of sexes in the table is number of female subjects with the remaining being defined as males, i.e., 12 female and 9 males in diet group and 14 women and 9 males in the diet-ex group. We have made changes to clarify this. The table 1 now read “Female subjects (male subject), n.”
In addition, we have made a clarification in the materials and method secretion that all women included in the study were indeed post-menopausal:
”Briefly, we included 32 overweight and obese (BMI 25-40 kg/m2) weight-stable (i.e., <5% weight loss) men and postmenopausal women with T2D.”
Another limit of the study is that some differences can be found with respect to baseline features for the two patient’s groups (i.e., age, insulin sensitivity indices) and this may have biased, at least to some extent the observed results. Also, some patients may have been more compliant than others with respect to paleolithic diet, standard recommendations for physical activities and even metformin use. As optimal matching is quite difficult to occur in a small, randomized study like this, I would suggest strengthening the need for larger studies to corroborate the present metabolomics data.
RESPONSE: We agree with the reviewer that baseline features may be of importance for interpretation of the results. We have added a comment on this in the discussion
“Another limitation of the study is potentially confounding effects of medications and differences in baseline insulin sensitivity, which should be considered in future studies.”
Regarding diet adherence, this comment overlaps with another reviewer’s comment. We have added a reference from a separate study of diet adherence from this cohort.
“Of note, a previous publication from our study group reported that the Paleolithic-type diet increased the dietary PUFA content, which was associated with lower levels of circulating TAGs independent of weight loss (ref 26). However, the study by Martensson et al (ref 26) could not conclude any difference in dietary adherence between the intervention groups.” Ref 26: Martensson et al , Nutrients 2021).”
The considerable amelioration of insulin sensitivity after only 12 weeks of paleolithic diet (independently of additional exercise) is unsurprising in view of recent evidences of a protective role for Mediterranean diet foods against insulin resistance. Please refer to doi: 10.3390/nu12041066 and doi: 10.3390/nu12103042 as suitable references.
RESPONSE: Thanks for pointing to this. We have added the reference and a comment linked to this in the discussion:
“Of relevance is that increases in dietary PUFA, such as in Mediterranean diets, have previously shown similar increases in insulin sensitivity after 12 weeks of diet (ref 27).”
Reviewer 3 Report
This manuscript by Chorell et al describes insulin sensitivity in both the plasma and liver in overweight type II diabetic women and links specific metabolomics and lipidomic signatures in plasma and skeletal muscle . This is the design included the ‘before and after’ implementation of a Paleolithic diet with or without exercise.
The results are interesting and advance our understanding of insulin sensitivity with a relatively novel approach utilizing these metabolic profiles and outcomes in insulin sensitivity. Thus, this information is definitely of interest and is sufficiently novel and unique to assure that it will be well received by the scientific community related to the study of insulin sensitivity.
I would offer a couple of recommendations that would improve the reporting and make information more focused. First of all the abstract should include a sentence as to the fact that all subjects [n=26] were female and were relatively obese ranging from 25 to 40 BMI. Related to that point table 1 has a couple of confusing entries. For instance, the first line relates to female subjects and describes n=12 and 14 for the two groups, but it’s not clear what percent [%] means when it says ‘nine’ after the 12 and after the 14. It certainly can’t be 9% of the total population of 26... that makes no sense. Also, lines 350 351 suggests that there were 13 subjects in each group, but table 1 shows 12 and 14 .please rectify.
Also further down in table 1 is the term ‘abdominal height’ in centimeters. What does that mean?.. is that the bellybutton height above the pubic bone :>) or some abstract measure that we don’t know about as a general readers of anatomy and physiology. …This should be clarified as a footnote if need be.
Also line 46 in the Introduction… the sentence structure is not good English. Why is the word ‘with’ inserted?.
Miscellaneous. Why were cereals excluded from the Paleolithic diet here? As the original Paleolithic diet certainly included wild grains. Do you mean commercially processed cereals?..so state if so.
I may have missed it, but I did not see any attempt to characterize the paleolithic diet relative to the basal diet in terms of their relative CHO:fat:PROt energy ratio. Even an estimate would be useful, and it would be relevant to include an estimate of the quality of fat as the SFA:MUFA:PUFA ratio because interesting differences were found in the TAG fatty acid profile of skeletal muscle. I suspect this has something to do with the fact that PUFA intake increased with the Paleolithic diet, which as the literature would suggest, improves overall energy metabolism, which, in turn, further allowed the exercise group to perform better in terms of fat oxidation and peripheral insulin sensitivity. [eg. see riseus and Willett 2009. Diet fat and risk of T2DM in women.]
Considering the wide range in the BMI, was there any attempt to bracket results from 25 to 30, 31 to 35, and 36-40 to determine whether there might be any trends in the data related to adiposity.?
Line 178.. Contains a typo ’insulin’.
Figure 3a. Y axis appears to be incorrectly labeled ‘rate of glucose disappearance’. Forgot the ‘glucose’.
Author Response
This manuscript by Chorell et al describes insulin sensitivity in both the plasma and liver in overweight type II diabetic women and links specific metabolomics and lipidomic signatures in plasma and skeletal muscle . This is the design included the ‘before and after’ implementation of a Paleolithic diet with or without exercise.
The results are interesting and advance our understanding of insulin sensitivity with a relatively novel approach utilizing these metabolic profiles and outcomes in insulin sensitivity. Thus, this information is definitely of interest and is sufficiently novel and unique to assure that it will be well received by the scientific community related to the study of insulin sensitivity.
I would offer a couple of recommendations that would improve the reporting and make information more focused. First of all the abstract should include a sentence as to the fact that all subjects [n=26] were female and were relatively obese ranging from 25 to 40 BMI. Related to that point table 1 has a couple of confusing entries. For instance, the first line relates to female subjects and describes n=12 and 14 for the two groups, but it’s not clear what percent [%] means when it says ‘nine’ after the 12 and after the 14. It certainly can’t be 9% of the total population of 26... that makes no sense. Also, lines 350 351 suggests that there were 13 subjects in each group, but table 1 shows 12 and 14 .please rectify.
RESPONSE: We thank the reviewer for highlighting that our presentation of sex-distribution in the different groups are misleading due to a percent sign, which is an error. The presented distribution of sexes in the table is number of female subjects with the remaining being defined as males, i.e., 12 female and 9 males in diet group and 14 women and 9 males in the diet-ex group. We have made changes to clarify this. The table 1 now read “Female subjects (male subject), n.”
According to reviewers comment we added “overweight and obese” to describe the included T2D in this study.
Also further down in table 1 is the term ‘abdominal height’ in centimeters. What does that mean?.. is that the bellybutton height above the pubic bone :>) or some abstract measure that we don’t know about as a general readers of anatomy and physiology. …This should be clarified as a footnote if need be.
RESPONSE: Thanks for pointing to this. We have made clarifications in the text accordingly. The term abdominal height or sagittal abdominal diameter is a well-established measure of abdominal adiposity with strong association with obesity-related diseases, as compared to waist circumference that can be quite difficult to measure in an obese individual. Accordingly, we changed the term abdominal height to sagittal abdominal diameter in table 1 and in the results section. In addition, we added a description and reference in the method section to clarify measure of this; “Sagittal abdominal diameter was measured as the height of the abdomen at the umbilical level when the subject was lying down on a flat surface with their legs straight (added a new reference num 29; Alderback et al J Obes 2021, DOI: 10.1155/2021/6647328).”
Also line 46 in the Introduction… the sentence structure is not good English. Why is the word ‘with’ inserted?.
RESPONSE: Thank so much for highlighting this error. We have removed “with” from this sentence.
Miscellaneous. Why were cereals excluded from the Paleolithic diet here? As the original Paleolithic diet certainly included wild grains. Do you mean commercially processed cereals?..so state if so.
RESPONSE: This comment highlights a very interesting and valid point. There are many paleolithic-type diets reported in the scientific literature and even more in the public. None of these provide an exact replica of what was eaten during the paleo era, since we do not know. Our motivation for excluding cereals from the diet were to make the diet easier to manage and increase coherence to the diet for the individuals. Also, today, cereals are often defined as breakfast cereals or bread as compared to non-processed/wild cereals, which were less likely the participant would include in their diet. Thus, we excluded all cereals, processed as well as unprocessed (wild). This is similar to other published paleolithic-type diets studies, such as Lindeberg et al Diabetologia 2007 (DOI: 10.1007/s00125-007-0716-y. ), Jonsson et al Cardiovascular Diabetol. 2009 (DOI: 10.1186/1475-2840-8-35) and Osterdahl et al in Eur J Clin Nutr. 2008 (DOI: 10.1038/sj.ejcn.1602790). We have not addressed this matter in more details since it is addressed in the base publications which is referred to in this paper (see reference 11 and 26) in row 335.
I may have missed it, but I did not see any attempt to characterize the paleolithic diet relative to the basal diet in terms of their relative CHO:fat:PROt energy ratio. Even an estimate would be useful, and it would be relevant to include an estimate of the quality of fat as the SFA:MUFA:PUFA ratio because interesting differences were found in the TAG fatty acid profile of skeletal muscle. I suspect this has something to do with the fact that PUFA intake increased with the Paleolithic diet, which as the literature would suggest, improves overall energy metabolism, which, in turn, further allowed the exercise group to perform better in terms of fat oxidation and peripheral insulin sensitivity. [eg. see riseus and Willett 2009. Diet fat and risk of T2DM in women.]
RESPONSE: We thank the reviewer for this valid comment. The intervention-induced changes in CHO, fats and protein are indeed of interest to the results interpretation. These data are well described in a previous publication (ref 11). The current study is a sub-study of this cohort. Numbers for the full study group were (CHO:FAT:PROT, E%):
DIET GROUP AT BASELINE (41:39:17)
DIET GROUP AT 12 WEEKS (31:42:24)DIET-EX AT BASELINE (42:34:18)
DIET-EX AT 12 WEEKS (27:45:2)
We can conclude that the participants in both intervention groups significantly reduced their CHO and saturated fat intake and increased their intake of protein and mono- and polyunsaturated fats.
In line with this we have included a sentence in the methods section that describe the alterations in dietary intake of CHO:FAT:PROT:
“A previous publication showed that the Paleolithic-type diet significantly altered the participants carbohydrate, protein and fat intake with a decreased carbohydrate and saturated fat intake and an increased intake of protein and mono- and polyunsaturated fats (ref 11).”
We also agree with the reviewer that the altered fat quality as well as adherence may have had an impact on the exercise intervention effect. The dietary adherence and fatty acid composition of the this Paleolithic-type diet have been investigated in a separate publication by Martensson et al. We have added a section in the discussion to address this:
“Of note, a previous publication from our study group reported that the Paleolithic-type diet increased the dietary PUFA content, which was associated with lower levels of circulating TAGs independent of weight loss (ref 26). However, the study by Martensson et al (ref 26) could not conclude any difference in dietary adherence between the intervention groups.” Ref 26: Martensson et al , Nutrients 2021, DOI: 10.3390/nu13030969).”
Considering the wide range in the BMI, was there any attempt to bracket results from 25 to 30, 31 to 35, and 36-40 to determine whether there might be any trends in the data related to adiposity.?
RESPONSE: Unfortunately, we do not have power to further examine the influence of adiposity on our results.
Line 178.. Contains a typo ’insulin’.
RESPONSE: This has been corrected.
Figure 3a. Y axis appears to be incorrectly labeled ‘rate of glucose disappearance’. Forgot the ‘glucose’.
RESPONSE: This has been corrected.